# Anaemia and Hypoproteinaemia in Pregnant Sheep during Anaesthesia

**DOI:** 10.3390/ani9040156

**Published:** 2019-04-11

**Authors:** Gabrielle C. Musk, Helen Kershaw, Matthew W. Kemp

**Affiliations:** 1Animal Care Services, University of Western Australia, Crawley, WA 6009, Australia; helen.kershaw@uwa.edu.au; 2Department of Obstetrics and Gynaecology, University of Wetsern Australia, Crawley, WA 6009, Australia; matthew.kemp@uwa.edu.au

**Keywords:** pregnant sheep, anaemia, anaesthesia

## Abstract

**Simple Summary:**

Anaemia during pregnancy is commonly reported in humans and is considered a significant health issue as it is associated with adverse neonatal outcomes including low birth weight, preterm birth and possible perinatal mortality. Pregnant sheep are a common model for research investigating the causes and consequences of preterm birth, but the incidence and significance of anaemia during pregnancy is not known. However, intraoperative anaemia in pregnant sheep has been reported previously. In addition, it has been demonstrated that the cause of intraoperative anaemia is not pregnancy alone as pregnant sheep in a farm environment did not develop anaemia. The aim of this study was to document the red blood cell and protein content of maternal blood before, during and after anaesthesia and surgery to elucidate the cause and duration of intraoperative anaemia in pregnant sheep in a research context. The concentration of red blood cells and protein decreased during anaesthesia and were attributed to the use of anaesthetic drugs and fluids as opposed to the pregnancy status of the sheep.

**Abstract:**

The aim of this study was to document the packed cell volume (PCV), haemoglobin concentration and total protein concentration of maternal blood before, during and after anaesthesia. Six singleton Merino-cross pregnant ewes at 116-117 days of gestation were premedicated with intramuscular acepromazine (0.02 mg/kg) and buprenorphine (0.01 mg/kg), and anaesthesia was induced with intravenous midazolam and ketamine. Anaesthesia was maintained with isoflurane in 100% oxygen. Serial blood samples were collected the day before anaesthesia (baseline), immediately prior to induction of anaesthesia (pre-op), at the end of the procedure (intra-op) and the following day (post-op). There was a significant change in the PCV during the study (*p* = 0.003) with an initial decrease of 12.5% from the baseline (0.36 (0.36–0.4) to 0.315 (0.29–0.34), *p* = 0.044), a further intraoperative decrease of 41.7% from the baseline (0.21 (0.195–0.245), *p* = 0.002) and an increase the day afterwards (0.3 (0.285–0.35), *p* > 0.99 compared with baseline). The haemoglobin concentration also changed (*p* < 0.0001) (baseline: 114 (111.8–123); pre-op: 97 (77.25–104.5), 14.9% decrease, *p* = 0.022; intra-op: 70 (61.5–83.25), 38.5% decrease, *p* = 0.0009; post-op: 101.5 (96.25–114) g/L, *p* > 0.99). Likewise the change in total protein during the study was significant (*p* = 0.0003) and decreased from the baseline [70 (67.25–70.75) g/L] prior to anaesthesia (61 (58.25–64.5) g/L, 12.9% decrease, *p* = 0.0437) and further during anaesthesia (55.5 (53.75–63.25) g/L, 20.7% decrease, *p* = 0.0021) with an increase [63 (61.25–67) g/L, *p* > 0.99] on the first post-op day. In conclusion, intraoperative anaemia and hypoproteinaemia occurred in this study. These alterations are attributed to a combination of the side effects of acepromazine and haemodilution.

## 1. Introduction

Anaemia during pregnancy is commonly reported in humans and is considered a significant health issue as it is associated with adverse neonatal outcomes including low birth weight, preterm birth and possible perinatal mortality [1]. Approximately 38% of pregnant women worldwide are anaemic as defined by a haemoglobin concentration < 110 g/L [2]. Pregnant sheep are a common model for research investigating the causes and consequences of preterm birth but the incidence and significance of anaemia during pregnancy is not known. However, intraoperative anaemia in pregnant sheep has been reported previously [3,4]. In addition, it has been demonstrated that the cause of intraoperative anaemia is not pregnancy alone as pregnant sheep in a farm environment did not develop anaemia [5].

The aim of this study was to document the packed cell volume (PCV), haemoglobin concentration and total protein concentration of maternal blood before, during and after anaesthesia and surgery to investigate the cause and duration of intraoperative anaemia in pregnant sheep in a research context.

## 2. Materials and Methods

The study was approved by the Animal Ethics Committee of the University of Western Australia (RA/3/100/1575) in accordance with the Code of Practice for the Care and use of Animals for Scientific Purposes [6]. The ewes were undergoing anaesthesia for instrumentation (maternal and fetal jugular catheterisation) for a pharmacokinetic project.

Six singleton Merino-cross pregnant ewes at 116–117 days of gestation (term ~150 days) were enrolled in the study. The sheep were sourced from a commercial farm in the south-west of Western Australia and introduced to the Large Animal Facility 2 to 3 weeks prior to surgical intervention. The indoor housing was in shared raised floor pens and then in single pens in the perioperative period. Sheep were fed pellets, oaten chaff and lupins with a mineral mix calculated on bodyweight and pregnancy status. Rooms were controlled for temperature (20.5–21.5 °C) and maintained on a 12:12 hour light:dark cycle.

The sheep were anaesthetised as previously described [4]. Food and water were not withheld prior to premedication. The ewes were premedicated with a combination of acepromazine (0.02 mg/kg, A.C.P. 2 Injection, 2 mg/mL, Ceva Delvet Pty Ltd, New South Wales, Australia) and buprenorphine (0.01 mg/kg, Temgesic, 0.3 mg/mL, Reckitt Benckiser, New South Wales, Australia) by intramuscular injection 30–40 minutes prior to induction of anaesthesia. Anaesthesia was induced with a combination of midazolam (0.25 mg/kg, Midazolam Injection, 5 mg/mL, Pfizer Australia Pty Ltd, NSW, Australia) and ketamine (5 mg/kg, Ketamil, 100 mg/mL, Troy Laboratories, New South Wales, Australia) by intravenous injection into a catheter placed in a cephalic vein, and the trachea was subsequently intubated (8.5 mm internal diameter, cuffed, Portex Ltd, England, UK). Anaesthesia was maintained with isoflurane (1–2%, Attane Isoflurane, Bayer Australia Ltd, New South Wales, Australia) in 100% oxygen delivered through a circle breathing system. The isoflurane vaporiser was adjusted as judged by an experienced veterinary anaesthetist to maintain an adequate depth of anaesthesia. Intermittent positive pressure ventilation was commenced immediately following endotracheal intubation to maintain normocapnia (end tidal CO_2_: 35–45 mmHg) using an Aestiva 5 anaesthetic machine with a Smartvent ventilator (Aestiva 5 with Smartvent 7900, Datex-Ohmeda, GE Healthcare, Gothenburg, Sweden). Volume controlled ventilation was utilised with a set tidal volume of 8 to 12 mL/kg, a respiratory rate between 8 and 12 breaths per minute and positive end-expiratory pressure of 5 cmH_2_O. Intravenous fluid therapy with 0.9% sodium chloride (NaCl) at 10 mL/kg/h was administered throughout anaesthesia. The animals were positioned in dorsal recumbency. Physiological monitoring with a multiparameter monitor included noninvasive blood pressure (oscillometric method) from a cuff around a dorsal pedal artery, oxyhaemoglobin saturation (SpO_2_, with the pulse oximeter probe positioned on the pinna or tongue) and side stream capnography (sample rate 150 ± 20 mL/min) (Surgivet V9203; Smiths Medical, Norwell, MA, USA). The ventilator variables were also recorded. The ventral abdomen and the neck of the ewe were clipped and prepared for aseptic surgery. A ventral midline laparotomy and hysterotomy were performed for placement of a central venous line in the foetus percutaneously. A maternal central venous catheter was also placed in a maternal jugular vein percutaneously.

Serial blood samples were collected on four occasions: the day before anaesthesia from the jugular vein or cephalic vein (baseline); immediately prior to induction of anaesthesia from the catheter placed in the cephalic vein (pre-op); at the end of the procedure from the maternal jugular catheter (intra-op); and the following day from the maternal jugular catheter (post-op). Blood was collected into EDTA anticoagulant and the analyses were performed by a commercial veterinary pathology laboratory (VetPath Laboratory Services, Ascot, Western Australia, Australia), to determine the PCV, haemoglobin concentration (Hb, g/L) and total protein concentration (TP, g/L).

For analgesia the ewes received buprenorphine prior to anaesthesia, ketamine during induction of anaesthesia, a line block for the ventral abdominal wound with ropivacaine (10 mL total volume, Naropin 1%, Astra Zeneca, New South Wales, Australia) and a transdermal fentanyl patch as previously described [7]. The ewes were monitored in the postoperative period to ensure they recovered from anaesthesia uneventfully. Wellbeing was monitored for four postoperative days, during which the pharmacokinetic study was performed. At the end of the study the ewes were euthanised with an intravenous injection of pentobarbitione (160 mg/kg).

Data were tested for normality and compared with analysis of variance (Freidman test) and Dunn’s multiple comparison test (GraphPad Prism, San Diego, CA, USA). A *p*-value < 0.05 was considered significant and the data are presented as mean (± SD) or median (25–75 percentile).

## 3. Results

The ewes weighed 56.7 (± 5.7) kg and anaesthesia time was 77.5 (± 2) minutes. Anaesthesia, surgery and recovery were uneventful in the six ewes. By pooling all the intraoperative data the mean arterial blood pressure was 50 (± 7.6) mmHg, the heart rate was 82 (± 17) beats per minutes, the oxyhaemoglobin saturation was 93 (± 3) %, the expired carbon dioxide was 41 (± 4) mmHg and the peak inspiratory pressure during mechanical ventilation was 17.2 (± 1.8) cmH2O.

There was a significant change in the PCV during the study (*p* = 0.003) with an initial decrease of 12.5% from the baseline (0.36 (0.36–0.4) to 0.315 (0.29–0.34), *p* = 0.044), a further intraoperative decrease of 41.7% from the baseline (0.21 (0.195–0.245), *p* = 0.002) and an increase the day afterwards (0.3 (0.285–0.35), *p* > 0.99 compared with baseline) (Figure 1). The pattern of change in the haemoglobin concentration was similar to that for the PCV and was also significant (*p* < 0.0001) (baseline: 114 (111.8–123); pre-op: 97 (77.25–104.5), 14.9% decrease, *p* = 0.022; intra-op: 70 (61.5–83.25), 38.5% decrease, *p* = 0.0009; post-op: 101.5 (96.25-114) g/L, *p* > 0.99) (Figure 2). Likewise the change in TP during the study was significant (*p* = 0.0003) and decreased from the baseline (70 (67.25–70.75) g/L) prior to anaesthesia (61 (58.25-64.5) g/L, 12.9% decrease, *p* = 0.0437) and further during anaesthesia (55.5 (53.75–63.25) g/L, 20.7% decrease, *p* = 0.0021) with an increase (63 (61.25–67) g/L, *p* > 0.99) on the first post-op day (Figure 3).

## 4. Discussion

Given the reported incidence of intraoperative anaemia in pregnant sheep the aim of this study was to determine the cause and duration of intraoperative anaemia in pregnant sheep in a research context. Intraoperative anaemia and hypoproteinaemia occurred in this study and partially resolved by the first postoperative day. While the values for each of the three measured parameters did not return to the baseline on the first postoperative day, there were no statistically significant differences between these post-op values and the baseline values.

From the baseline values there was a decrease in both the PCV and haemoglobin concentration to a level consistent with anaemia [2]. The results of this study are comparable to previous studies where intraoperative anaemia occurred in singleton and twin pregnancies with reported values for the PCV of 19.6 (± 2.6) [3] and 22.4 (± 2.6) [4]. In addition intraoperative haemoglobin concentrations in pregnant sheep have been reported to be 76.24 (± 8.6) g/L approximately 30 min after induction of anaesthesia and 67.4 (± 11.7) g/L 30 minutes later [4]. These reported values are comparable to the results of this study. Previous studies in sheep have not measured the PCV between the time of premedication and induction of anaesthesia and in this case the initial drop from the baseline to the pre-op value is attributed to acepromazine. Acepromazine produces significant haematological side effects in animals, decreasing the PCV by 20% to 30% within 30 min of administration for at least 2 h [8]. In this study the greatest decrease in PCV occurred after approximately 70 minutes of anaesthesia, but the initial decrease was 12.5%, suggesting that acepromazine contributes to intraoperative anaemia.

The mechanism of anaemia following the administration of acepromazine has been demonstrated in various species including sheep to be splenic relaxation and consequent increase of red cells held there [9,10,11]. This theory has been investigated in horses and dogs, whereby the alpha adrenoreceptor antagonism of acepromazine causes vasodilation, splenic relaxation and erythrocyte storage [10,12]. The duration of effect of acepromazine on the PCV is considered to be dose-dependent as splenic storage capacity will plateau with higher doses, but persists for longer [12]. In horses, doses of 0.05 mg/kg by intramuscular injection have effects on the PCV for 12 hours [12]. Translating these horse data to a pregnant sheep model receiving 0.02 mg/kg of acepromazine by intramuscular injection is difficult as the impact of pregnancy on the capacity of splenic engorgement is not known in this species. Nevertheless, the relatively low dose of acepromazine administered to the sheep in this study was likely to be associated with at least the initial decrease in PCV (and Hb concentration) and the incomplete (although not statistically different) increase to the baseline values on the first postoperative day.

The decrease in the PCV and haemoglobin concentration during anaesthesia and surgery (from pre-op to intra-op) is not likely to be an effect from the acepromazine alone. Normovolaemic haemodilution is a result of progressive exchange of the blood with crystalloids or colloids and the physiological responses to haemodilutional anaemia are similar between species [13]. In our study, where the pattern of change in all three parameters was similar, intraoperative haemodilution exacerbated the initial decrease in the PCV, Hb and TP concentration. The administration of an isotonic crystalloid during anaesthesia in this study represents standard practice in our laboratory in an effort to offset the hypotensive effects of drugs like acepromazine [8] and isoflurane [8]. Normovolaemic haemodilution occurs intraoperatively as a result of vasodilation, haemorrhage and intravenous fluid therapy [14]. In this study it is likely that haemodilution contributed to the progressive decline in the PCV, Hb and TP concentrations intraoperatively. Volume status was not measured directly in this study, nor is it routinely measured during anaesthesia of pregnant sheep in our facility. Indirect assessment of blood volume can be made by simultaneously measuring arterial blood pressure and central venous pressure during anaesthesia. Accurate measurement of these parameters is invasive and in this study a noninvasive technique for arterial blood pressure measurement was utilised. The inherent inaccuracy of this technique translated to numbers consistent with hypotension so it is feasible that the sheep in this study were not normovolaemic but had a relative hypovolaemia with haemodilution.

The manifestations of intraoperative anaemia in pregnant sheep were not apparent in this or other studies in our facility, as there were no overt signs of postoperative organ injury and mortality. This tolerance of intraoperative anaemia is attributed to its relatively short duration being associated with anaesthesia, surgery and the capacity of compensatory mechanisms to maintain oxygen delivery to tissues [14]. Nevertheless acute haemodilution presents a risk to animals and in humans it is associated with an increased risk of stroke, renal injury, mycocardial ischaemia and mortality [14]. The physiological responses to haemodilutional anaemia are similar between species and the objective of compensatory mechanisms is to prevent tissue hypoxia [13]. Observed cardiovascular responses are described as a progressive increase in cardiac index, heart rate and stroke volume and a decrease in systemic vascular resistance [14]. In a rat study, severe normovolaemic haemodilution was tolerated for 150 minutes without a significant increase in lactate concentration [13]. Extrapolation to pregnant sheep seems reasonable but in future studies intraoperative monitoring of arterial lactate concentrations will assist in the decision making process for management of blood volume. Additionally, invasive blood pressure measurement should also be considered for pregnant sheep undergoing anaesthesia surgery so accurate information can be used to determine the most appropriate fluid therapy regime. Comprehensive physiological monitoring of arterial blood pressure as previously mentioned and heart rate should be standard. As haemodilutional anaemia presents a risk for tissue hypoxia, which may be particularly relevant in pregnant animals where adequate oxygenation of the placenta and foetus is essential, and efforts to evaluate intraoperative tissue oxygen delivery should be made. These efforts are largely undefined at the present time but there is promise around the value of nitric oxide mediated methaemoglobin production that may be used as a biomarker of anaemic stress [14].

The parallel alterations in haemoglobin concentration in this study are directly related to the changes in the PCV. Given that the haemoglobin protein is contained with red cells, the fewer circulating red cells the lower the circulating haemoglobin concentration. Erroneous causes of low haemoglobin concentration include dilution of red blood cells due to vasodilation and intravenous fluid therapy [15], but, in this study, the simultaneous decrease in both the PCV and haemoglobin concentration are considered directly linked.

The dose of acepromazine used in this study may be considered as low at 0.02 mg/kg by intramuscular injection. Despite this dose the side effects of acepromazine were apparent as previously discussed. However, the noninvasive blood pressure measurement technique is unlikely to be accurate as the sheep were in dorsal recumbency and the position of the cuff was higher than the right atrium, the site of the cuff was not clipped of wool and other authors have demonstrated that other oscillometric devices underestimate the mean arterial blood pressure [16]. Despite the inaccuracy of noninvasive blood pressure measurement, it remains useful insofar as trends or changes over time can be evaluated and there is a low risk of morbidity associated with instrumentation with these devices [16].

The changes in total protein concentration were similar to those for the PCV and haemoglobin concentration. This similarity suggests the cause of intraoperative hypoproteinaemia was also haemodilution. Decreasing total protein concentrations are reported in dogs during anaesthesia and laparotomy attributed to anaesthetic drug administration, dilutional effects of intravenous fluid therapy and loss of plasma protein into the abdominal cavity [9]. In another study in dogs, the TP decreased following anaesthesia and surgery to an extent to that reported herein [17].

There are a number of limitations to this study, which must be considered when interpreting the results. The study is small and it was an opportunistic study so sample collection occurred at times that did not interfere with the primary objectives of the study proper. Likewise, a control group of nonpregnant sheep could not be included. To understand the side effects of acepromazine invasive blood pressure measurement would have been a better way to evaluate blood pressure. Finally additional samples after the first postoperative day would have provided more information about recovery from the decreased PCV, haemoglobin concentration and TP concentrations.

## 5. Conclusions

In conclusion, intraoperative anaemia and hypoproteinaemia occurred in the sheep in this descriptive study. The extent of these alterations was statistically significant and is attributed to a combination of the side effects of acepromazine and haemodilution.

## Figures and Tables

**Figure 1 animals-09-00156-f001:**
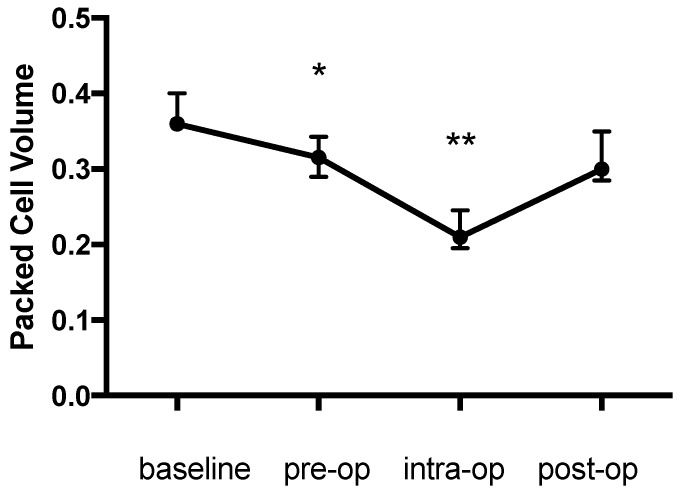
Packed cell volume during the study period. * *p* = 0.044 pre-op compared to baseline; ** *p* = 0.002 intra-op compared to baseline.

**Figure 2 animals-09-00156-f002:**
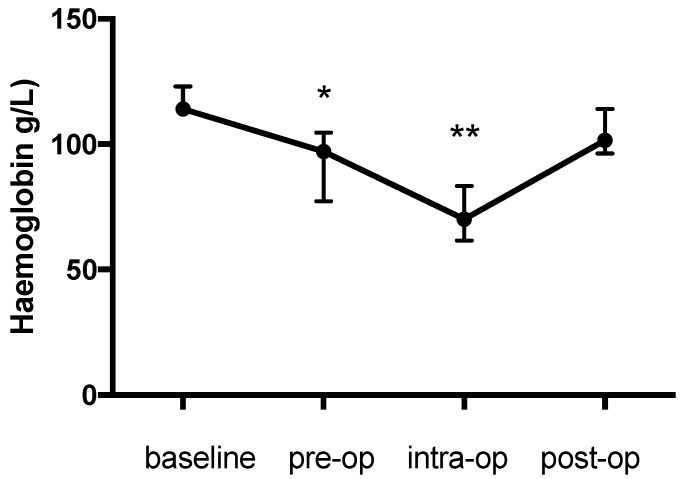
Haemoglobin concentration (g/L) during the study period. * *p* = 0.022 pre-op compared to baseline; ** *p* = 0.0009 intra-op compared to baseline.

**Figure 3 animals-09-00156-f003:**
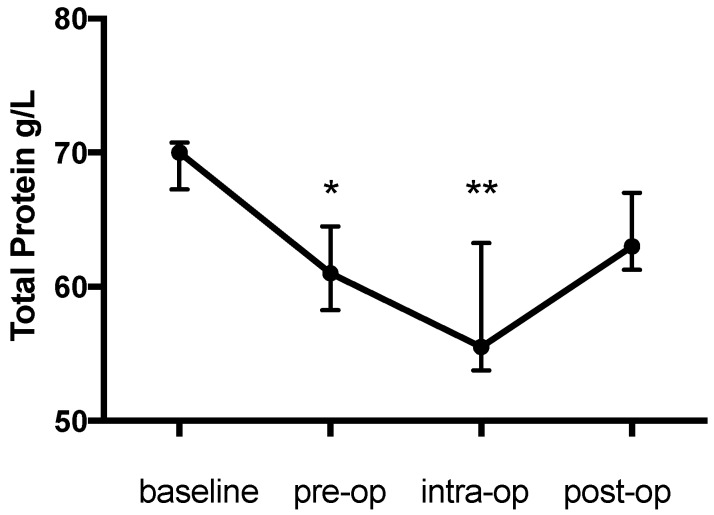
Total protein concentration (g/L) during the study period. * *p* = 0.0437 pre-op compared to baseline; ** *p* = 0.0021 intra-op compared to baseline.

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
