# Peer review of "Anaemia and Hypoproteinaemia in Pregnant Sheep during Anaesthesia"

_animals, 2019, doi:10.3390/ani9040156_

Round 1
Reviewer 1 Report
This is an interesting study well described. I have a few minor suggestions:
Lines 16,17 'the cause ....is not due to...' Either 'the cause is' or 'is due to', not both.
Line 21 'was' should be replaced with 'were'.
Line 33 'compared with', not 'compared to'
Line 52 Either 'the cause is' or 'is due to', not both.
Line 56 Would 'investigate the cause' better describe the aim? It seems that while valid suggestions of the cause of maternal PCV, Hb and TP have been made the cause has not been established definitively.
Line 80 remove '1mg/ml'
Line 124 'compared with', not 'compared to'
Author Response
Reviewer 1
This is an interesting study well described. I have a few minor suggestions:
Lines 16,17 'the cause ....is not due to...' Either 'the cause is' or 'is due to', not both.
This change has been made as suggested
Line 21 'was' should be replaced with 'were'.
This change has been made as suggested
Line 33 'compared with', not 'compared to'
This change has been made as suggested
Line 52 Either 'the cause is' or 'is due to', not both.
This change has been made as suggested
Line 56 Would 'investigate the cause' better describe the aim? It seems that while valid suggestions of the cause of maternal PCV, Hb and TP have been made the cause has not been established definitively.
This change has been made as suggested
Line 80 remove '1mg/ml'
This change has been made as suggested
Line 124 'compared with', not 'compared to'
This change has been made as suggested
Reviewer 2 Report
Anaemia and hypoproteinaemia in pregnant sheep during anaesthesia
This study sets out to assess blood cell parameters during anaesthesia in a pregnant sheep model, to better understand intra-op anaemia in pregnant sheep.
This is an excellently written and very clear paper, reporting on a study of specialist but well defined interest. I have not much to add. See specific comments below.
L 90 please provide details (brand etc) for blood pressure cuff, pulse oximeter probe and side stream capnography probe unless they are all part of the Surgivet V9203 system in which case this should be clear from the wording – however I cannot see a blood pressure cuff and oximeter on the list of included items at the Smith Medical website.
L 237 ff It could be worth pointing out as a further limitation of the study is the lack of a non-pregnant control group to assess whether pregnancy plays a role in the observed effects of anaesthesia. As an opportunistic study this may well have not been an option and hence this is not a criticism but I feel it should be pointed out that firm conclusions on whether pregnancy may play a role here cannot be drawn.
Author Response
This study sets out to assess blood cell parameters during anaesthesia in a pregnant sheep model, to better understand intra-op anaemia in pregnant sheep.
This is an excellently written and very clear paper, reporting on a study of specialist but well defined interest. I have not much to add. See specific comments below.
L 90 please provide details (brand etc) for blood pressure cuff, pulse oximeter probe and side stream capnography probe unless they are all part of the Surgivet V9203 system in which case this should be clear from the wording – however I cannot see a blood pressure cuff and oximeter on the list of included items at the Smith Medical website.
This sentence has been reworded to make it clearer that all the monitoring was using one unit – the surgivet.
L 237 ff It could be worth pointing out as a further limitation of the study is the lack of a non-pregnant control group to assess whether pregnancy plays a role in the observed effects of anaesthesia. As an opportunistic study this may well have not been an option and hence this is not a criticism but I feel it should be pointed out that firm conclusions on whether pregnancy may play a role here cannot be drawn.
This additional limitation has been included as requested.